# Fecal Cortisol Metabolites Indicate Increased Stress Levels in Horses During Breaking-In: A Pilot Study

**DOI:** 10.3390/ani15121693

**Published:** 2025-06-07

**Authors:** Julia Krieber, Aurelia C. Nowak, Jakob Geissberger, Oliver Illichmann, Sabine Macho-Maschler, Rupert Palme, Franziska Dengler

**Affiliations:** 1Institute of Physiology and Pathophysiology, Department of Biological Sciences and Pathobiology, University of Veterinary Medicine Vienna, 1210 Vienna, Austriaaurelia.nowak@vetmeduni.ac.at (A.C.N.);; 2Experimental Endocrinology, Department of Biological Sciences and Pathobiology, University of Veterinary Medicine Vienna, 1210 Vienna, Austria; sabine.macho-maschler@vetmeduni.ac.at; 3Department of Livestock Tissue Metabolism, Institute of Animal Sciences, University of Hohenheim, 70599 Stuttgart, Germany

**Keywords:** equine, stress, glucocorticoids, feces, training, riding, equestrian sports

## Abstract

Horses are instinctively flight-oriented animals and thus highly sensitive to external influences. Sport horses have to deal with a lot of different and potentially stressful situations and the criticism from animal welfare organizations against equestrian sport is rising. This pilot study aimed to investigate how much stress young horses experience during initial training, when the transition from an unridden horse to a riding horse takes place. To determine the stress level, we measured fecal cortisol metabolites (FCMs) of young and unridden horses and of horses in various training states between less than one year to over three years in training. We could not only show that FCM levels of unridden horses were lower than those of horses in training, but also that FCMs were highest in horses that were newly broken-in. After reaching a peak within the first year under the saddle, more experienced horses tended to have lower FCMs. This leads to the conclusion that initial training is a stressful period for the horses and therefore particularly careful handling during this sensitive phase of life is important for their future career as a sport horse.

## 1. Introduction

Horses have been a means of transportation and warfare, as well as companions, for humans for centuries, and the popularity of horseback riding as a leisure activity continues. In the past decades, horses have gained importance as partners, and their welfare is increasingly being considered. Nonetheless, stress is a daily companion in the life of many horses, especially when they are in training, taking part in competitions or being transported [1,2,3,4]; but also common diseases such as lameness and external influences, e.g., changes in the weather, are important stressors for horses [5]. “Stress” is a vague term, and many researchers have already tried to find the ideal definition. According to Ramos and Mormède, “stress is considered as the response of an organism to environmental stimuli (stressors) which threaten its internal equilibrium, also called homeostasis” [6]. Fraser et al. propose that “an animal is said to be in a state of stress if it is required to make abnormal or extreme adjustments in its physiology or behavior in order to cope with adverse aspects of its environment and management” [7]. Both definitions suggest that stress is an uncomfortable condition and highlight the importance of avoiding stressful situations to protect the physical and mental wellbeing of horses. It must be differentiated between eustress and distress, i.e., “positive” or “negative” stress. While eustress induces a transient activation of the hypothalamo–pituitary–adrenal axis and results in a successful adaptation, distress exceeding the animal’s adaptation potential results in a prolonged activation of stress pathways and potentially harm to health and welfare [8,9]. Besides the adaptations in the physiology and behavior of the animals that are necessary to deal with stress, it could be shown that particularly transport and lameness can cause reduced pulmonary defense mechanisms and also have negative effects on other immunological functions [5], and thus impair the horses’ health. Therefore, in the interest of animal welfare and health, it is important to identify triggers potentially activating the stress response in horses.

So far, stress is usually measured by assessing clinical (heart rate, body weight, and activity level) and endocrinological (plasma cortisol, catecholamine, and beta-endorphin concentrations) parameters [5]. Due to the fact that stress leads to increased glucocorticoid secretion by increasing the activity of the hypothalamo–pituitary–adrenal axis [10], there are already numerous studies that identified assumingly stressful situations for horses by means of increases in cortisol release. Besides typical situations like training [1,11], transport [3,12,13,14], competitions [2,15,16], veterinary examinations [17,18], social instability [19], and inadequate management and housing conditions [20,21,22], all of which have been shown to increase plasma cortisol levels, breaking-in, i.e., the transition from a young, unridden horse to a riding horse and sport partner for humans, can also be seen as a stressful situation for the animal. Animal welfare organizations are increasingly criticizing equestrian sports for the potentially negative impact it may have on horses, particularly equine athletes at a high-performance level and young horses during breaking-in. However, there are only few studies that investigate the actual stress level of horses during initial training. Schmidt et al. identified particularly the mounting of a rider as a stressor for 3-year-old sport horses [4] and demonstrated a quick adaptation to the new situation, while Pilger et al. reported that also the pretraining of young stallions leads to an initial stress response [23]. Both studies determined changes in salivary cortisol concentrations, heart rate, and heart rate variability and thus had to manipulate the animals repeatedly, which could additionally have stressed the young horses. In contrast, Gorgasser et al. did not find any increase in adrenocortical activity in young Quarter horses within the first month of training using fecal cortisol metabolites (FCMs) as indicator and thus concluded that the animals were able to deal well with their new tasks [24]. All of these studies investigated stress levels in the horses only during the training period, neglecting the possibility of generally increased baseline values due to the onset of training, and a comparison over a longer period of time, e.g., between horses that are not trained at all, more experienced riding horses, and horses during initial training, is lacking. Therefore, from the current state of the literature it cannot be concluded whether the transition phase is particularly stressful for horses or not. In this pilot study, we aimed to examine if the stress levels in horses during breaking-in differ from those in horses that are not ridden or more experienced horses in training.

While many studies utilized plasma or saliva cortisol concentrations [25,26,27] as a marker, we chose to determine FCMs in order to avoid additionally stressing the animals by venipuncture or taking saliva samples. Non-invasive techniques to measure glucocorticoid metabolites in feces have been developed for several species [10,28,29]. Also, in horses an 11-oxoaetiocholanolone enzyme immunoassay [30] was proven to reliably quantify a group of FCMs without the need for manipulating the animal [31,32,33]. Several studies have already demonstrated that increased plasma cortisol concentrations aligned with increased FCM levels in horses exposed to potential stressors like transport and veterinary examinations [2,14,15].

## 2. Materials and Methods

### 2.1. Horses

We included a total of 41 horses from one stud farm, aged between 1 and 20 years, in this study. According to their level of training, they were categorized into two groups, “unridden” and “in training”.

The group of unridden horses (*n* = 28) was composed of female, male intact and castrated warmblood horses, Haflingers, and a Noriker. They were housed in open stables with seasonal paddocks. Their barns were strewn with straw, equipped with a deep litter system, and were usually cleaned every two weeks, depending on the amount of manure. Water was provided ad libitum and the horses were fed with hay, haylage, and grass. The animals had contact with humans from birth and were halter trained, but were not trained otherwise before breaking-in. This group was further subdivided in three different age groups of “1–2 years old” (*n* = 6), “2–3 years old” (*n* = 10), and “3–4 years old” (*n* = 12).

The group of horses in training contained *n* = 13 warmblood geldings, which were kept individually in boxes (3 m × 3 m) with straw bedding and were mucked out daily. They were fed with hay and oats and had ad libitum access to water. Their training involved two to three sessions of approximately 30 min of riding or free jumping per week, with a few small individual modifications. This group was further classified, based on the time the horses had been under the saddle, into “ridden for 3–6 months” (*n* = 6, all 4 years old), “ridden for 1 year” (*n* = 3, 4–5 years old), and “ridden for 3 years or longer” (*n* = 4, aged 8–20 years).

### 2.2. Sample Collection and Processing

The study was conducted at the beginning of May 2023. A fresh fecal sample consisting of the inner part of several horse apples of every horse was collected in the morning. The samples of the ridden horses were collected on the day after training, and all samples were immediately stored at −20 °C. Training took also place in the morning, so that all samples were taken approximately 22 ± 2 h after training.

The extraction of FCMs was performed as described by Merl et al. [34]. In brief, the whole sample was mixed by hand and 0.5 g of feces from different parts were mixed with 5 mL 80% methanol (Merck KGaA, Darmstadt, Germany), vortexed for 30 min, and centrifuged at 2500× *g* for 15 min (AllegraTM X-12R, Beckman Coulter, Krefeld, Germany). A total of 1 mL of the supernatant was transferred into new vials and mixed with 0.25 mL 5% NaHCO_3_ and 5 mL diethylether (Merck KGaA, Darmstadt, Germany) before spinning down again and freezing at −20 °C overnight. The next day, the supernatant was transferred into new vials and dried down before being redissolved in 0.5 mL assay buffer. Concentrations of 11,17-dioxoandrostanes were measured via an 11-oxoetiocholanolone enzyme immunoassay, as described before [30]. This method has been successfully validated for use in horses [31]. Each sample was measured in duplicate and the mean value of all technical replicates was used for statistical analysis.

### 2.3. Statistical Analysis

For statistical analysis, FCM values were compared using Sigma Plot 14.5 (Systat Software Inc., Düsseldorf, Germany) with either a Mann–Whitney rank sum test for comparison of two groups or a one way ANOVA on ranks with a post hoc Holm–Sidak test for the comparison of more than two groups. The data were tested for normality using a Shapiro–Wilk test (*p* < 0.05) and for equal variance with a Brown–Forsythe test (*p* > 0.05). Correlation analysis was performed using a Pearson product moment correlation. Significance was assumed at *p* < 0.05.

## 3. Results

### 3.1. FCM Concentrations Are Higher in Horses in Training

FCM concentrations among the unridden horses ranged from 4.5 to 14.0 ng/g feces (median 7.3 ng/g), while FCM levels of the horses in training were between 7.5 and 18.1 ng/g feces (median 11.6 ng/g). There was a statistically significant difference (*p* < 0.001, Mann–Whitney rank sum test) between the two groups (Figure 1).

### 3.2. FCM Concentrations Are Higher Directly After Breaking-In

There was no difference regarding the FCM concentrations between the different age groups of unridden horses (Figure 2). Within the group of “1–2 years old” horses, a range from 5.6 to 13.0 ng/g feces (median 6.6 ng/g) was measured. The values of the “2–3 years old” horses were between 5.2 and 9.1 ng/g feces (median 7.3 ng/g) and the “3–4 years old” horses had levels from 4.5 to 14.0 ng/g feces (median 7.7 ng/g).

In contrast, horses which had been in training for less than one year had significantly higher FCM concentrations, ranging from 7.5 to 18.1 ng/g feces (median 13.1 ng/g) compared to unridden horses (*p* < 0.01, one way ANOVA + post hoc Holm–Sidak test). FCMs in horses that had been in training for at least one year no longer differed significantly from those in the group of unridden horses and showed a tendency to have lower FCMs in relation to the time under the saddle. The groups “ridden for one year“ and “ridden for 3 years or longer” were combined for statistical analysis due to their small group sizes, and their median FCM concentrations ranged from 6.9 to 16.2 ng/g feces (median 8.8 ng/g; Figure 2 and Table 1).

### 3.3. FCM Concentrations Show a Weak Correlation to Training Status

Pearson product moment correlation indicated a weak positive correlation (r = 0.440, *p* < 0.01) between FCM levels and the time the horses had been under the saddle at the time of measurement (Figure 3).

## 4. Discussion

Animal welfare in equestrian sport is increasingly under debate. A common point of criticism concerns the breaking-in of the horses, i.e., getting them used to the saddle, bridle, and rider. However, it is hard to objectively judge or even quantify the strain that is actually induced in the animals by this procedure. The aim of this pilot study was to investigate whether horses in training, and particularly those that have recently been broken-in, have higher stress levels compared to unridden horses. FCM concentrations in fecal samples of young and unridden horses of different ages and of horses with various training statuses were compared. It is important to highlight that we did not only measure FCMs during the period of breaking-in but also compared them across horses of various training stages, spanning years before and after initial training and thereby providing a comprehensive investigation of FCM levels in horses throughout different phases of training. Additionally, in our study the horses were all from the same farm and not transported to another facility and the measurements took place in their familiar environment, thus avoiding a bias that is present in many other studies.

The FCM values we observed, particularly in the unridden horses, align well with baseline values measured with the same method in another study using a larger horse population [35], supporting the validity of the measurements and indicating the baseline character of these measurements. We showed that the median FCM concentrations of unridden horses were lower than those of horses in training. Furthermore, our results suggest that after reaching a peak within the first year under the saddle, FCM levels are lower in more experienced horses. Interesting findings were not only the higher median concentration of FCMs in horses in training, but also a larger range of variation in these horses, although the group of ridden horses was more homogenous regarding sex and breed than that of the unridden horses. This might be interpreted as a higher stress level in horses that are being ridden [32], with an individual variability of their ability to deal with that stress. This should not be disregarded, as individual horses may have different coping mechanisms in order to deal with excessive strain that can lead to unwanted behavior [36]. The variability could also be caused by different riders, whose level of experience as well as the arousal experienced by the rider can influence the horse’s behavior and trigger the animal’s flight response [36]. It is therefore important to ensure that training is carried out by an experienced, confident, and calm person, especially in the early stages of training when the horse is already exposed to several new stimuli. Unfortunately, detailed information on the riders was not available for this study, but all of them were experienced in training (young) horses.

Our findings support previous reports that breaking-in is a stressful period of time for young horses. Schmidt et al. demonstrated that equestrian training is challenging for the horses, especially in the beginning, by measuring salivary cortisol concentrations, heart rate, and heart rate variability [4]. Conversely, a study performed in young Quarter horses did not show any increase in FCMs during the first month of breaking-in [24]. These conflicting results may be caused by different breeds, housing conditions, and/or training style. Although training horses without a rider through activities like free movement, jumping obstacles, lunging, or using a horse walker can already elevate the stress response [23], it is important to highlight that the mounting of the rider is a very sensitive step when it comes to the process of breaking-in [4]. In the wild, there is no natural equivalent to a human riding on a horse’s back, except the scenario of a predator catching its prey. Therefore, to make equitation even possible, a high tolerance from the young horses towards their rider is needed [37], who is not only on their back but also out of their field of vision [38]. Observing and understanding the individual reaction of horses to new and potentially stressful situations (e.g., the first mounting of the rider) could not only be helpful to select them for different intended purposes, according to their ability to cope with this kind of stress [39], but is also relevant to avoid undesirable, “negative” behavior and resulting accidents [40]. However, it must also be mentioned that the highest stress levels observed both in our and other studies during initial training do not reach those induced by transport, pain, or excitement due to unusual events [34,35], indicating that the training process might cause some distress in the animals but no serious impairment of animal welfare.

It is important to highlight that as soon as the young horses are used to their rider, initial training is rather a physical exercise than a mental task [4]. However, it should be borne in mind that physical exercise can also lead to an increased stress response [41]. Nonetheless, it is noteworthy that in our study horses with a longer training history exhibited lower FCM levels. This could either indicate that physical activity alone is not the primary driver of elevated FCM values, as experienced horses also undergo intensive training requiring substantial strength and fitness. Alternatively, the higher FCM concentrations in less experienced horses may reflect the greater physical and psychological demands associated with initial training, which may elicit an increased stress response until the animals become habituated. To maintain the horses’ will to cooperate with humans, it is essential to know that handling and training can influence them either in a positive or a negative way [42], and that negative experiences are able to delete positive ones from their memory [43]. For these reasons, it is even more relevant to be careful during the initial training of young horses.

It has been reported that horses kept on pastures and in groups are able to deal better with new training situations [44] and show less stereotypies and abnormal stress-related behavior than horses kept individually in boxes [45]. Our findings also suggest that keeping young horses in open stables in groups has a positive impact, as they were already being accustomed to human contact and halter training without showing a corresponding increase in FCM values. This implies that exposure to human interaction and basic handling does not induce stress per se, further underlining that the primary triggers occur during the process of breaking-in. However, since this was also the time point when the horses’ environment changed from being kept in an open stable in groups to being stabled in individual boxes for the first time, it is not possible to differentiate precisely which experience is the main reason for the rise in FCM concentrations in our study and whether keeping them in groups during the process of breaking-in would have attenuated the stress response. However, other studies suggest no or rather an inverse impact of group housing on the horses’ resilience during breaking-in. Schmidt et al. investigated two groups, one of which included only mares housed in group stables and one including stallions that were kept in individual boxes. In both groups salivary cortisol levels increased during breaking-in and even more in the group of mares than in the stallions, emphasizing that this probably has a greater influence on the animals than the housing conditions [4]. Similarly, Pilger et al. reported that group housing had no stress-reducing effects during pretraining [23].

It must not be neglected that this pilot study has several limitations, and further studies are warranted to confirm our results. The main limitation of our study is that we could only measure one sample per animal and thus we could not perform a longitudinal observation of FCMs in individual horses before, during, and after breaking-in to identify different factors influencing FCM levels. Additionally, only a small number of horses were enrolled in the study, and including a larger sample of horses of the same breed, thereby accounting for variations in temperament and aptitude, would have been desirable to minimize individual variability. In particular, the group of experienced horses was very small due to the pilot character of our study, and the significance of these results is therefore limited. More detailed information on the horses’ fitness level, training schedule, and health status should be recorded in future studies. Further studies including long-term observations of the same pasture-kept horses without any changes in housing and social groups while breaking-in and under more standardized training conditions would be interesting and should be performed in the future. In addition, it would be helpful to combine different methods to detect and quantify stressful events in horses, for example, salivary cortisol concentrations, heart rate measurement, and behavioral analysis. Determining salivary cortisol may be a better method to detect increases in cortisol levels induced by specific events throughout the day, whereas FCMs appear with a delay of 24 h after a stressful event in the feces because of the intestinal passage of the horse [28]. FCMs may thus be more representative of the general wellbeing of a horse within the last 24 h, and in animals with a long passage time such as horses, circadian rhythms do not affect FCMs as much as plasma or salivary cortisol [29]. Also, hair samples could be included to measure cortisol levels. However, although recommended by many researchers as a measure of chronic stress [46], it remains still unclear what period is reflected in those values [47], which could however partly be overcome by applying a shave–reshave technique [48]. In contrast, it is easier to identify a specific stressor in salivary or plasma samples taken before and after the event. However, when measuring plasma cortisol, the sampling process itself is stressful for the horses when it comes to repeated venipuncture or fixation of the head, particularly when the horses did not have a lot of contact with humans before. Therefore, we chose a non-invasive method using fecal samples in this study.

## 5. Conclusions

In conclusion, our preliminary results show that the FCM concentrations of unridden horses were significantly lower than those of horses in training. The larger range of variability in the FCM levels of ridden horses suggests individual variations regarding their ability to deal with stress. Another interesting finding was that FCM levels reach a peak within the first year of being ridden, confirming that breaking-in is a critical time point for young horses.

## Figures and Tables

**Figure 1 animals-15-01693-f001:**
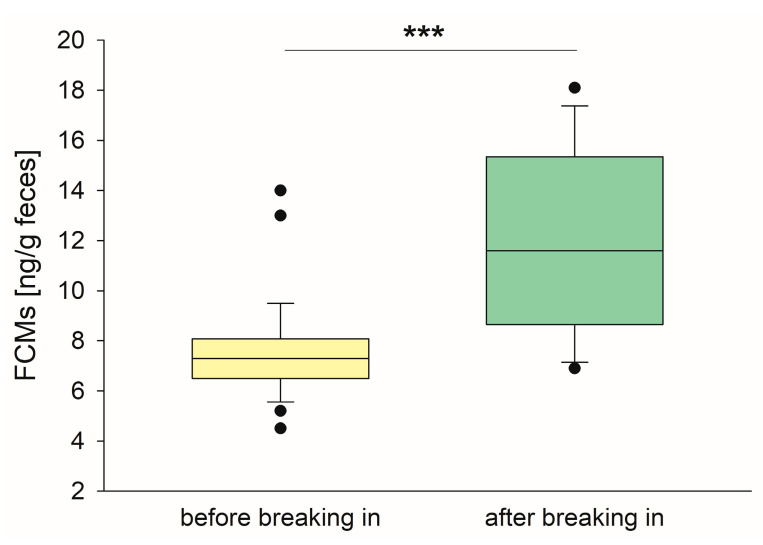
FCM concentrations of unridden horses (yellow box, *n* = 13) were significantly lower than of horses in training (green box, *n* = 28). Boxplots represent median FCM concentrations (line) in the different age groups ± 25th and 75th percentiles (box) and with the 10th and 90th percentiles shown (whiskers). Outliers are marked by dots. *** *p* < 0.001, Mann–Whitney rank sum test.

**Figure 2 animals-15-01693-f002:**
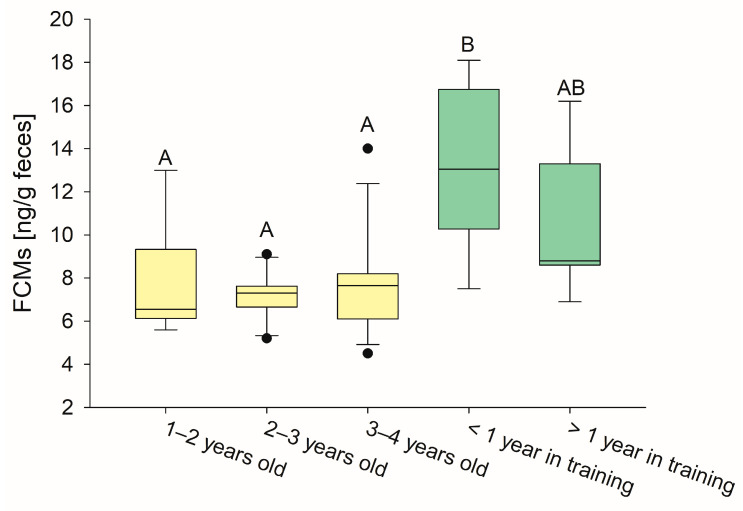
Newly broken-in horses had higher FCM concentrations compared to unridden horses. Horses that have been ridden for less than one year had significantly higher FCM levels than horses that were not yet broken-in. After more than one year in training there was no statistical difference between FCM levels in horses in training (green boxes) and unridden horses (yellow boxes). Boxplots represent median FCM concentrations (line) in the different age groups ± 25th and 75th percentiles (box) and 10th and 90th percentiles (whiskers). Outliers are indicated by dots. Different letters indicate significant differences between groups, *p* < 0.01, one way ANOVA + post hoc Holm–Sidak test.

**Figure 3 animals-15-01693-f003:**
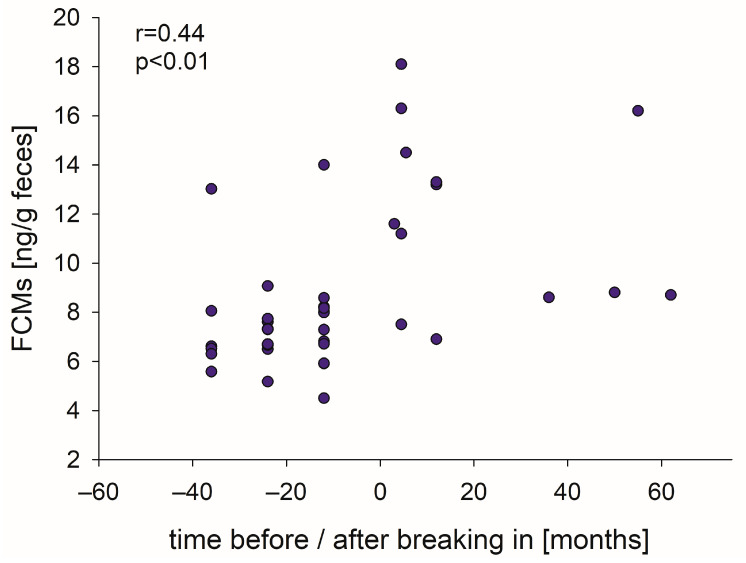
Relation between FCM levels of individual horses and the time (months) before and after breaking-in, respectively.

**Table 1 animals-15-01693-t001:** Median and quartiles of FCM concentrations in ng/g feces in the different groups of horses. Different superscripts indicate significant differences between groups, *p* < 0.05, one way ANOVA + post hoc Holm–Sidak test.

Group	Median	75%	25%
Unridden, 1–2 years old	6.6 ^a^	9.3	6.1
Unridden, 2–3 years old	7.3 ^a^	7.6	6.7
Unridden, 3–4 years old	7.7 ^a^	8.2	6.1
In training for 3–6 months	13.1 ^b^	16.8	10.3
In training for 1 year	13.2 ^ab^	13.3	6.9
In training for >3 years	8.8 ^ab^	14.4	8.6

## Data Availability

Raw data generated within this study are available from the corresponding author upon reasonable request.

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
