# Peer review of "Fecal Cortisol Metabolites Indicate Increased Stress Levels in Horses During Breaking-In: A Pilot Study"

_animals, 2025, doi:10.3390/ani15121693_

Round 1

Reviewer 1 Report

Comments and Suggestions for Authors

Lines 54-55-56. This idea is a bit confusing.  "Both definitions lead to the conclusion 
that stress is an uncomfortable condition and that it is essential, for the physical and mental wellbeing of horses, to avoid stressful situations." . Stress is important to avoid stress is what i am understanding and I am not sure that is what the authors want to state...could improve writting of the idea.

Lines 126-127-128 Was there an exact amount of hours after the training that the feces were collected? or it was "whenever"? The amount of time that passes between the "stressful" event and feces collection is important. How much of the sample was collected, was it all and then homogenized? was it avoiding the soil? only 100g? please add that information.

Line 174: states: in compared to. Fix appropiately.

Lines 282-283-284 "The main limitation of our study is that we could only measure one sample per animal and thus we could not perform a longitudinal observation of FCMs in individual 
horses before, during, and after breaking in to identify different factors influencing FCM 
levels."  This part worries me a bit since you are not stating a standard amount of time from which the stressful situation happened and the samples were taken. Circadian rhythm may influence FCM and introduce some noise to your study if the samples were not taken at consistent times, it would be great if you could include a reference about circadian rhythm not affecting FCM as much as cortisol in blood to account for that.

In general, well written, nicely discussed. Very interesting topic.

Author Response

Please see file attached.

Reviewer 2 Report

Comments and Suggestions for Authors

Thank you for letting me review this article. The article is well written, and the topic is relevant for the equestrian world and need further research.

However, in my opinion, the article has two main limitations: a) the small sample size; b) the collection of only one sample per horse. The small sample size did not allow a reliable statistical analysis, thus I would suggest to the authors considering to limit the inferential statistics to larger groups, where the sample size support reliable analysis. In addition, as the authors themselves has underlined in the discussion section, the collection of only one fecal sample per horse give an idea of the stress condition of only the previous 24 hours. This is not sufficient to have an idea of the overall stress status of the horse, and it cannot support robust conclusions.

Therefore, I would suggest increasing the number of animals and/or collect more fecal samples over time. In addition, could be interesting to consider other animal welfare indexes, such as behaviour or cortisol measurement in hair/mane, which could be useful in the measurement of chronic stress.

In conclusion, I would suggest reconsidering the statistical analysis and data collection, or considering to submit the world not as full article but as short communication/preliminary work.

Author Response

Please see file attached.

Reviewer 3 Report

Comments and Suggestions for Authors

I would like to thank the authors for submitting this manuscript. However, it contains some inconsistencies and weaknesses, particularly in the methodology and the presentation of results. In several sections, the authors seem to suggest that the study supports broader conclusions than what the data actually allow.

Additionally, I recommend a thorough revision of the English throughout the manuscript. There are incorrect word choices (e.g., line 12 – "flight animals") and some sentences that are difficult to understand (e.g., lines 18–20).

I also point out that the manuscript does not include an ethics committee approval code. Since animals were used, such approval is essential.

Line 110: Please specify the number of females, males, and animals from each breed.
This raises a major concern. Different breeds have different temperaments and aptitudes for work. Having non-homogeneous groups in a study like this represents a significant limitation.

The "ridden horses" group is highly fragmented. To serve as a reliable comparison group, it should include a higher number of individuals in each phase of the work.

Line 122 – "small individual modifications": Please specify what these modifications were.

There is a lack of detail throughout the entire manuscript.
Given that the study aims to assess the influence of exercise, a prior medical examination should have been conducted (musculoskeletal or dental issues can significantly affect stress during exercise). A body condition score assessment should have been included, and a standardized exercise intensity chart should have been used (including average heart rate; percentage of work at walk, trot, and canter; total work duration, etc.).

Line 127: "...collected in the day after the training" – this needs to be more specific.
If one sample was collected 12 hours after exercise and another 6 hours after, this could influence FCM measurement. Additionally, under what conditions did the horses exercise? What time of day?
Exercise performed at 12:00 p.m. in high temperatures versus 8:00 a.m. in cooler conditions can influence FCM levels.

Statistics
Shapiro–Wilk should be applied first. Mann–Whitney is a non-parametric test, while ANOVA is a parametric test—they differ not only in the number of groups compared, but also in the assumptions underlying the data. This distinction must be made clearly.

Lines 201–203: Caution—this statement is not accurate and contradicts what is written in lines 282 and 284. A comprehensive investigation would require following the same horses over the different stages. This was not done. The study collected only one sample per horse, from different horses that were in different “athletic states.” Each horse has individual personality traits and physical condition, which introduces variability that must be acknowledged.

Line 227: No, it doesn't! To make such a statement or claim this result, it would be necessary to compare the FCM measurement with another well-established or validated method and demonstrate comparable results. This statement should be removed.

Lines 250–260: Again, fitness level, coordination, and core muscle strength development can all influence stress levels and are not discussed in the manuscript. For example, a dressage horse may experience increased stress around the age of 7–8 when beginning collection movements, which are physically more demanding.

A thorough final paragraph discussing the study’s limitations is essential. These include:

Different breeds (with different temperaments and aptitudes);

Very small group sizes;

Lack of detailed information about type of work and fitness level;

No assessment of the animals' health status;

As mentioned in the introduction, “external factors” play a significant role in stress levels and were not considered (e.g., environmental temperature).

I must strongly disagree with the classification of salivary sampling via swab as invasive. Even young horses typically accept this method without difficulty.

Lines 75–85: This section reads more like a discussion than an introduction. Please either remove it or move it to the discussion section.

Comments on the Quality of English Language

 I recommend a thorough revision of the English throughout the manuscript. There are incorrect word choices (e.g., line 12 – "flight animals") and some sentences that are difficult to understand (e.g., lines 18–20).

Author Response

Please see file attached.

Round 2

Reviewer 2 Report

Comments and Suggestions for Authors

Thank you for revising the manuscript as suggested. The changes have significantly improved it.

I would suggest adding a reference in line 74 after "inadequate management and housing conditions".

In addition, I do not fully agree with the statement regarding hair cortisol in line 308: "it remains still unclear what period is reflected." The use of a shave–reshave technique allows for an estimation of the hair growth period and could provide an indication of the time frame over which cortisol has accumulated.

Author Response

Thank you for revising the manuscript as suggested. The changes have significantly improved it.

I would suggest adding a reference in line 74 after "inadequate management and housing conditions".

A: Thank you for the suggestion, we included references in line 74.

In addition, I do not fully agree with the statement regarding hair cortisol in line 308: "it remains still unclear what period is reflected." The use of a shave–reshave technique allows for an estimation of the hair growth period and could provide an indication of the time frame over which cortisol has accumulated.

A: Thank you for this valuable comment. We added this thought to the discussion in lines 313f.

We would like to thank the reviewer again for the time and effort invested and for helping us to improve the manuscript!

Reviewer 3 Report

Comments and Suggestions for Authors

I would like to thank the authors for their attempt to respond to my comments and improve the manuscript, but in my opinion, this manuscritp does not have sufficient quality and robustness to be published in this journal. 

Author Response

I would like to thank the authors for their attempt to respond to my comments and improve the manuscript, but in my opinion, this manuscritp does not have sufficient quality and robustness to be published in this journal. 

A: We would like to thank the reviewer for their suggestions in the last round. We believe their contributions significantly improved the manuscript, and we regret that you do not share this view. We kindly ask the reviewer to provide a more detailed description of the remaining shortcomings, so that we may address them appropriately.
